# Mediterranean Diet Influence on SARS-CoV-2 Vaccine Adverse Reaction: Friend or Foe?

**DOI:** 10.3390/nu16121846

**Published:** 2024-06-12

**Authors:** Paola Gualtieri, Giulia Frank, Rossella Cianci, Antonella Smeriglio, Angela Alibrandi, Laura Di Renzo, Domenico Trombetta

**Affiliations:** 1Section of Clinical Nutrition and Nutrigenomic, Department of Biomedicine and Prevention, University of Tor Vergata, Via Montpellier 1, 00133 Rome, Italy; paola.gualtieri@uniroma2.it (P.G.); laura.di.renzo@uniroma2.it (L.D.R.); 2PhD School of Applied Medical-Surgical Sciences, University of Tor Vergata, Via Montpellier 1, 00133 Rome, Italy; giulia.frank@ymail.com; 3School of Specialization in Food Science, University of Tor Vergata, Via Montpellier 1, 00133 Rome, Italy; 4Department of Translational Medicine and Surgery, Catholic University of the Sacred Heart, 00168 Rome, Italy; 5Fondazione Policlinico Universitario A. Gemelli, Istituto di Ricovero e Cura a Carattere Scientifico (IRCCS), 00168 Rome, Italy; 6Department of Chemical, Biological, Pharmaceutical and Environmental Sciences, University of Messina, 98122 Messina, Italy; asmeriglio@unime.it (A.S.); dtrombetta@unime.it (D.T.); 7Department of Economics, University of Messina, 98100 Messina, Italy; angela.alibrandi@unime.it

**Keywords:** Mediterranean diet, sex differences, vaccine side effects, SARS-CoV-2

## Abstract

Background: The Mediterranean Diet (MedDiet) has long been recognized for its health-promoting attributes, with proven benefits in preventing cardiovascular and metabolic diseases. During the global COVID-19 pandemic, MedDiet’s potential to mitigate the impact of SARS-CoV-2 infection gained attention. This study aims to investigate the interplay among MedDiet adherence, immune system response to SARS-CoV-2 vaccines, and potential sex-related variations. Methods: A retrospective observational study was conducted through collecting data from a web survey for the Italian population. Adherence to the MedDiet was assessed using the Mediterranean Diet Adherence Screener (MEDAS); in addition, COVID-19 symptoms and vaccination details were also obtained. Results: Significant associations between MedDiet adherence, COVID-19 symptoms, and vaccine-related side effects were observed. Notably, females demonstrated distinct responses, reporting lymph node enlargement and a different prevalence and severity of vaccine side effects compared to males. Conclusions: This study highlights the protective role of the MedDiet against COVID-19 and emphasizes the relevance of sex-specific responses in vaccination outcomes according to MEDAS score.

## 1. Introduction

It is well known that adherence to a proper diet, such as the Mediterranean Diet (MedDiet), contributes to maintaining optimal health by providing the body with adequate preventive and curative support. This was also demonstrated in the case of COVID-19 [1]. The MedDiet is a dietary pattern of Mediterranean Sea countries based on the traditional foods and drinks consumed by these populations. The MedDiet is characterized by a high consumption of vegetables, fruits, legumes, unrefined grains, and extra virgin olive oil, a moderate consumption of dairy products and fats, and a low consumption of meat, processed foods, and sweets [2]. High adherence to the MedDiet has been shown to reduce the risk of mortality linked to cardiovascular [3] and metabolic diseases [4], such as type 2 diabetes [5], through a reduction in abdominal fat [6,7]. These effects are due to the biologically active components and macro- and micronutrients in foods in which the MedDiet is rich [8]. The benefits of the MedDiet are linked to its high polyphenol content that presents an anti-inflammatory, antioxidant, and immunomodulatory effect on inflammatory diseases [2,9]. Fruits and vegetables provide polyphenols that act as antioxidants by reducing oxidative stress and inflammation through the blocking of pro-inflammatory cytokines [10]. Omega-3 fatty acids contained in high amounts in fish also enable cell membrane protection by preventing the oxidation of polyunsaturated fatty acids and reducing Tumor Necrosis Factor α (TNF-α) levels [11]. In addition, the MedDiet has been found to reduce the cytokine levels involved in the proinflammatory response, such as C-reactive protein (CRP), Interleukin (IL)-6, and IL-1 [12]. 

After the binding between membrane angiotensin-converting enzyme 2 (ACE 2) and SARS-CoV-2 spike S-glycoproteins, viruses enter the cells and start the immune response to the infection by increasing proinflammatory cytokines, such as TNF-α, IL-6, IL-7, IL-8, Il-9, IL-10, Interferon gamma (IFN-γ), vascular endothelial growth factor (VEGF), and CRP, and elevating the production of reactive oxygen species (ROS) [13]. SARS-CoV-2 is able to elicit a strong cytokine storm, a consequent dysregulation of the immune system with the development of systemic inflammation and multiorgan failure [13]. During this inflammatory response, the MedDiet could provide beneficial effects on COVID-19 outcomes through its anti-inflammatory properties [8,14].

Several scientific studies have demonstrated a bidirectional interaction between lifestyle, immunity, and infections. Indeed, the immune response is compromised by an unhealthy lifestyle, unbalanced and inadequate eating habits, and the presence of disease states (obesity, cardiovascular disease, diabetes, neurodegenerative diseases, and cancer) [15]. Sex differences have also been reported in infectious diseases, including COVID-19 [16]. Indeed, women are more susceptible to autoimmune disorders and men are more prone to infectious diseases, potentially due to X chromosome protection and hormonal effects. Women often show stronger immune responses during viral infections, attributed to sex steroids, while higher ACE-2 expression in women may play a role in the less severe outcomes of COVID-19 [16]. However, whether sex has an influence, in synergy with the MedDiet, on the immune response to SARS-CoV-2 vaccines remains poorly investigated.

The aim of the present study is to investigate the relationship between MedDiet adherence and immune system response to SARS-CoV-2 vaccines, focusing on possible sex differences in a specific age group of non-pathological and drug-therapy-free subjects.

## 2. Materials and Methods

This retrospective observational population-based study, initiated in April 2023 by the Section of Clinical Nutrition and Nutrigenomic at the University of Rome Tor Vergata, employed a web survey conducted from 24 April to 4 September 2023, to collect data on sex, adherence to the MedDiet, and the side effects of COVID-19 vaccination among a specific age group of non-pathological and drug-therapy-free internet-confident subjects belonging to the Italian population. The survey, administered following the snowball model through online institutional platforms and social networks, adhered to inclusion and exclusion criteria, encompassing participants over 18 years old with internet access and excluding participants with psychotic disorders, pregnancy, breastfeeding, neoplastic diseases under treatment, drug use, and HIV/AIDS. The termination date of the survey was determined upon reaching a predefined minimum number of participants, calculated to attain a statistical power exceeding 80%. The questionnaire, available in Appendix A, was created utilizing Google Forms, accessible on any Internet-connected device. The questionnaire, consisting of 45 questions across five sections, covered self-reported personal data, anthropometrics, MedDiet adherence, COVID-19 experiences, and vaccination details. Adherence to the MedDiet was assessed using the validated Mediterranean Diet Adherence Screener (MEDAS) [17], categorizing participants into low (0–5 points), medium (6–9 points), and high adherence (≥10 points) and calculating differences in the compliance rates for each item.

The completion of the SARS-CoV-2 vaccination cycle was determined upon the administration of the required doses, as stipulated by Italian law.

The research adhered to national and international regulations and the Declaration of Helsinki (revised in 2013). The participants were thoroughly briefed on the study’s requirements and required to accept the data sharing and privacy policy following the EU GDPR No. 2016/679 before engaging in the study. The respondents submitted their responses through the Google platform, ensuring the anonymization of personal information to preserve their privacy. The survey’s anonymous nature precludes the tracing of sensitive personal data, exempting it from Ethics Committee approval [18]. Following completion, each questionnaire was transmitted to the Google platform, and the final dataset was downloaded in the form of a Microsoft Excel spreadsheet.

### Statistical Analysis

Data are represented as absolute frequencies and percentages in parentheses (%) for categorical variables or median and interquartile range in square brackets [IQR] for continuous variables. Variable distribution was assessed using the Shapiro–Wilk test, revealing a skewed distribution for all variables, therefore, the non-parametric approach was adopted for all statistical analyses. The Spearman correlation test was employed to examine the correlation among continuous variables. Differences in the MEDAS score variable between women and men were determined using Mann–Whitney U-test for independent samples. The Chi-square test of independence was utilized to explore associations between categorical variables, categorizing the questionnaire responses into ‘YES’ and ‘NO’, ‘Women’ and ‘Men’, or MEDAS classes. Additionally, binary and multinomial logistic regression models were estimated to investigate the associations between dependent categorical outcome variables and independent continuous or categorical variables. The following confounding factors were included: age, gender, BMI, and lifestyle (sedentary/active). Statistical significance was set at *p* < 0.05, and the analysis was performed using SPSS ver. 21.0 (IBM, Chicago, IL, USA).

## 3. Results

Overall, 776 participants responded to the survey. Specifically, 71,26% were women (aged 42.8 ± 18.36; BMI 23.77 ± 4.91) and 28,74% were men (aged 43.84 ± 16.93; BMI 25.76 ± 3.7). Significant sex differences were observed in weight (*p* < 0.0001), height (*p* < 0.0001), and BMI (*p* < 0.0001). No other sex differences were reported. A comparison of general and anthropometric characteristics is shown in Table 1.

Regarding the MEDAS group classification, 9.4% of women showed low adherence to the MedDiet, 62.57% medium adherence, and 28.03% high adherence. Of the men, 14.35% showed low adherence, 61.43% medium adherence, and 24.22% high adherence. Significant sex differences were observed in olive oil (*p* < 0.007), vegetables (*p* < 0.0001), red meat (*p* < 0.011), wine (*p* < 0.0001), legumes (*p* < 0.014), and white meat (*p* < 0.0001) consumption. No other sex differences were reported. Positive responses to the MEDAS by individual items and by adherence groups are shown in Table 2 and Figure A1 and Figure A2 (Appendix B).

The Mann–Whitney U-test for independent samples reported a significant difference in the MEDAS score between those who had or had not contracted COVID-19 (*p* < 0.0001). Particularly, those who developed COVID-19 (35.2%–34.9% of women and 35.9% of men) had a mean MEDAS score of 7.31 ± 2.5, while those who did not (64.8%) had a MEDAS score of 8.5 ± 1.85.

Logistic regression showed that a higher MEDAS score was associated with decreased COVID-19 symptoms (*p* < 0.0001; OR = 3.13; 95%CI 2.232–4.388), but not with decreased vaccination side effects before (*p* = 0.545; OR = 1.22; 95%CI 0.952–1.098) and after (*p* = 0.831; OR = 1.008; 95%CI 0.937–1.085) the vaccination cycle (Table 3).

The Chi-square test was performed to assess possible associations, both at the start and the end of the vaccination cycle, between individual side effects, MEDAS adherence, and sexes, respectively (Table 4 and Table 5, respectively). 

Specifically, no significant associations were observed between side effects and MEDAS groups (Table 4) and between side effects and sex, neither before (respectively, *p* = 0.56 and *p* = 0.82) nor after (respectively, *p* = 0.19 and *p* = 0.51) the vaccine cycle (Table 5).

Regarding individual side effects and MEDAS groups, a significant association was reported between MEDAS groups and tiredness at the beginning of the vaccine cycle (*p* < 0.001). No other significant associations were observed at either the beginning or the end of the vaccine cycle (Table 6).

Concerning women, a significant association was observed between MEDAS groups and lymph node enlargement at the end of the vaccine cycle (*p* < 0.021). No other significant associations were observed for women either at the beginning or at the end of the vaccine cycle (Table 7).

Concerning men, no significant associations were observed for males at either the beginning or end of the vaccine cycle (Table 8).

## 4. Discussion

In this study, a significant difference in the MEDAS scores between those who had or had not contracted COVID-19 was reported. Also, a significant correlation between MEDAS score and COVID-19 symptoms was shown. A significant association was observed between MEDAS groups and tiredness at the beginning of the vaccine cycle. Notably, women reported a significant association between MEDAS groups and lymph node enlargement at the end of the vaccine cycle. COVID-19 is considered to be a global pandemic, beginning in December 2019 and affecting not only the lives of people, but global health and socioeconomic systems [19]. The World Health Organization (WHO) estimated 774,395,593 reported cases worldwide until January 2024 [20], and 18.2 million deaths have been reported worldwide due to the COVID-19 pandemic [21]. Concerning the risk of SARS-CoV-2 infection and mortality, women are known to be less affected than men [22,23]. Men with COVID-19 are known to have higher mortality rates than women, possibly due to higher rates of comorbidities related to male norms, such as smoking and alcohol consumption [23]. In addition, men have also demonstrated prolonged viral shedding, lower cure rates, and a greater disease severity, likely influenced by hormonal factors and ACE-2 expression in the testis [22]. While susceptibility to disease does not differ significantly between genders, biological factors such as innate immunity and hormone levels may contribute to these disparities [22,23]. Our study showed that men had an infection rate of +1% compared with women.

There are several risk factors for COVID-19, including dietary style and lifestyle [21]. Indeed, an increased risk of severe COVID-19 has been observed in patients following the Western diet, which provides a high intake of saturated fats, sugars, and refined carbohydrates [24]. This dietary pattern is known to dysregulate the immune response and impair the host’s defense against viruses [25], stimulating the expression of pro-inflammatory cytokines leading to cytokine storm [26]. In addition, obese subjects have been shown to have a 46.0% increased risk of COVID-19 positivity [27]. In contrast, the MedDiet shows anti-inflammatory characteristics linked to the high consumption of foods rich in vitamins, minerals, polyphenols, and antioxidants [28], which can improve the immune response [29,30] and reduce the expression of pro-inflammatory molecules [31]. It is well known that the MedDiet leads to a significant reduction in circulating markers of oxidative stress and inflammation, such as ox-LDL [29] and CRP [31] levels, and a significant down-regulation of pro-inflammatory genes, such as CCL5, UCP2, BCL2, IRAK1, and DUOX2 [29].

Our study reported that a high MedDiet adherence had an effect on reducing the incidence of COVID-19. These results have been confirmed by several studies. It was observed that high adherence to the MedDiet is associated with more than a 60% decreased risk of COVID-19 [32]. Perez-Araluce et al. [33] showed that the MedDiet could provide potential protection against COVID-19, thus, as adherence to the MedDiet increases, a reduction in the risk of developing the disease is reported. Ponzo et al. [34] noted that the risk of SARS-CoV-2 infection was significantly associated with a lower MedDiet adherence. The MedDiet’s anti-inflammatory and immunomodulatory properties are well known [35]; therefore, it could present a protective effect against COVID-19 [14]. Indeed, its richness in phenolic compounds has not only anti-inflammatory, antithrombotic, and antioxidant effects, but also the capability to prevent the entry of SARS-CoV-2 through the Spike protein [36,37]. The MedDiet, renowned for its preventive effects on cardiovascular disease and type 2 diabetes, offers potential benefits in mitigating the severity of COVID-19 infection [38]. Rich in plant-based components, the MedDiet provides bioactive polyphenols. Particular flavonoids are known for their antioxidant and anti-inflammatory properties and can counteract the exaggerated inflammatory response associated with severe COVID-19 disease, potentially improving outcomes [38]. In addition, MedDiet components, such as nuts and dried fruits, offer antioxidant and anti-inflammatory molecules that promote cardiometabolic health [38].

Our study reported a significant association between MEDAS score and COVID-19 symptoms. Several studies have shown that high adherence to the MedDiet is associated with a lower risk of hospitalization following infection and a reduced severity of symptoms [33,39,40]. Particularly, the MEDAS score is inversely associated with COVID-19 symptoms [39]. It has also been shown that COVID-19 patients with high MedDiet adherence had lower serum CRP levels and erythrocyte sedimentation rates, and were 77% less likely to have severe COVID-19 [39]. The MedDiet’s features, such as a higher consumption of fruit, vegetables, and dietary fiber, have previously been associated with shorter hospitalization and recovery periods, lower serum CRP levels, and lower odds of having severe COVID-19 [40]. Micronutrients, in which the MedDiet is rich, have also been shown to reduce COVID-19 severity by reducing the risk of intensive care admission and shorter hospitalization periods [12].

To date, for the first time, we observed in our study that increased adherence to the MedDiet reduces the development of the tiredness side effect after the SARS-CoV-2 vaccine. To our knowledge, only one previous study has focused on how the side effects of SARS-CoV-2 vaccination can be improved by supplementation. Particularly, it was observed that ω-3, vitamins, and minerals supplementation can reduce the side effects of SARS-CoV-2 vaccines, such as diarrhea and nausea [41]. Thus, our results seem to confirm the role that specific nutrients, plentifully represented in the MedDiet, play in modulating the immune response [13]. 

Sex differences in SARS-CoV-2 vaccination side effects are well-documented. Gualtieri et al. [41] observed decreased side effects with prebiotics, probiotics, ω-3, and L-glutamine in men, while in women, a reduction was reported with vitamin D supplementation [18]. Self-reported side effects’ analyses post-SARS-CoV-2 vaccination consistently show a higher prevalence and severity in women compared to men for some side effects [42,43,44,45]. This trend persists across age groups, suggesting a robust sex dimorphism in vaccination-related adverse events and indicating a potential biological basis for sex-specific responses. Notably, Khalil et al. [42] reported a higher symptom prevalence in women, particularly for nausea. Duijster et al. [43] emphasized pronounced sex differences in injection site inflammation and nausea, with women being about twice as likely as men to develop these. Nishino et al. [44] reported in women more vaccine-related lymphadenopathy than that in men. In our study, women reported a rate of +0.67% of side effects compared with men at the beginning of the vaccination cycle. However, at the end of the vaccination schedule, women reported a −2.14% rate compared with men.

Only a few studies have shown that the MedDiet may influence the development of some side effects. Specifically, in a study of 24 participants undergoing chemotherapy treatment, a significant association between the MedDiet and nausea and vomiting was revealed [46]. The MedDiet showed a link to a reduction in the incidence and severity of nausea, abdominal pain, and bloating. The antioxidant and anti-inflammatory properties of the MedDiet were identified as potential factors in attenuating the pathways associated with nausea [47]. 

However, to our knowledge, the novelty of our report is the association between women and a higher MEDAS score with decreased COVID-19 symptoms and lymph node enlargement at the end of the vaccine cycle. The MedDiet is able to regulate gene expression related to oxidative stress; at the same time, sex exerts regulation on gene expressions linked to inflammation and oxidative stress [48]. Moreover, it is well established that there are sex differences in the immune response. Women present stronger inflammatory, antiviral, and humoral immune responses than men, due to stronger immune responses to vaccination compared to men [31]. This disparity is influenced by the binding of estrogens to specific receptors on specific immune cells [49]. Estrogens affect the signaling pathways involved in chemokine and cytokine production, resulting in a more robust immune response [50]. In contrast, testosterone and dihydrotestosterone tend to suppress the activity of immune cells, reducing the vaccination response and depressing the cytokine response [51]. Additionally, the X chromosome, which is present in women, harbors more genes related to immunity than the Y chromosome in men. Polymorphisms in these genes contribute to sex differences in immune responses, observed both before and reproductive age [50]. Several studies have also shown that immune status is associated with adherence to the MedDiet [52,53]. A study conducted by Bédard et al. reported that the impact of the MedDiet on systemic inflammation was comparable between women and men, with the individual’s overall inflammatory status influencing these effects specifically in men [31]. Different expressions between the sexes of genes involved in oxidative stress and inflammation have been reported as an effect of the MedDiet [48]. 

Our study has some limitations. The first is its relatively small sample size, which needs expansion to enhance heterogeneity, encompassing diverse populations beyond Italy. In addition, due to the retrospective study design, recall biases and causality associations cannot be excluded. Another constraint is the higher representation of women (71.26%) among the survey participants. This sex imbalance can potentially be attributed to their increased engagement with social media and greater online presence, particularly in seeking health-related information, as reported by the National Institute of Statistics [54], compared to men [55]. 

A notable strength of our study lies in the unbiased selection of the sample population facilitated by the widespread adoption of digital channels in Italy. With over 50 million internet users and nearly 44 million social media users in 2023, representing 86.1% and 74.5% of the Italian population, respectively, digital platforms provide a broad and accessible means of reaching potential participants [56]. Additionally, the presence of 78.19 million cellular mobile connections, exceeding the Italian population, highlights the extensive use of mobile technology, further enhancing the reach and inclusivity of our recruitment efforts [56]. Additionally, our investigation pioneers the exploration of the relationship between vaccine side effects, sex differences, and MedDiet adherence.

## 5. Conclusions

In conclusion, this study explains the pivotal role of the MedDiet not only in reducing the risk and severity of COVID-19, but also in influencing sex-specific responses to SARS-CoV-2 vaccines. The observed associations between MedDiet adherence, reduced COVID-19 symptoms, and specific vaccine side effects highlight the potential of personalized dietary interventions in healthcare. Understanding how dietary patterns interact with the immune system and vaccination outcomes opens up avenues for the application of the 4Ps Medicine. These findings underscore the significance of tailoring health recommendations based on individual dietary habits, offering a promising approach to enhancing vaccine responses and overall well-being. Integrating personalized dietary considerations into public health strategies may pave the way for more effective preventive measures and personalized healthcare interventions in the context of infectious diseases and vaccinations.

Based on our evidence, future studies could explore sex differences in the long-term effects of MedDiet adherence on COVID-19 outcomes and vaccination responses in different populations. Simultaneously, clinical trial studies should be conducted to investigate the underlying mechanisms by which MedDiet components modulate immune responses to viral infections and vaccination, with a focus on sex-specific differences. These investigations will provide valuable insights into the potential of MedDiet interventions in enhancing vaccine responses and reducing the severity of COVID-19. In addition, expanding research efforts to include larger and more diverse populations beyond Italy through population-based studies will clarify the generalizability of findings and take into account cultural and dietary variations.

## Figures and Tables

**Table 1 nutrients-16-01846-t001:** Comparison of participants’ general characteristics and anthropometrics.

	Whole Sample (*n* = 776)	Women (*n* = 553)	Men (*n* = 223)	*p*-Value
Age (years)	43.1 ± 17.96	42.8 ± 18.36	43.84 ± 16.93	^a^ 0.46
Weight (kg)	69.31 ± 15.44	64.53 ± 13.62	81.15 ± 13.13	^a^ 0.0001 ***
Height (cm)	168.42 ± 8.63	164.81 ± 6.46	177.38 ± 6.53	^a^ 0.0001 ***
BMI (kg/m^2^)	24.35 ± 4.69	23.77 ± 4.91	25.76 ± 3.7	^a^ 0.0001 ***
Lifestyle				^b^ 0.63
Sedentary	313 (40.3%)	226 (40.9%)	87 (39%)	-
Active	463 (59.7%)	327 (59.1%)	136 (61%)	-
Affected by COVID-19	273 (35.2%)	193 (34.9%)	80 (35.8%)	^b^ 0.93
Vaccinated for COVID-19	776 (100%)	553 (100%)	223 (100%)	^b^ 1.0
Groups of vaccinated				^b^ 0.47
One dose	11 (1.4%)	7 (1.2%)	4 (1.8%)	-
Two doses	139 (17.9%)	94 (16.9%)	45 (20.2%)	-
Three doses	626 (80.7%)	452 (81.9%)	174 (78%)	-

Values are expressed as mean and standard deviation (M ± SD) for continuous variables or as absolute frequency and percentage (*n* (%)) for categorical variables. ^a^ An independent samples non parametric Mann–Whitney U test was performed to determine sex differences. ^b^ A Chi-square test was performed to determine sex associations. Abbreviations: BMI, body mass index. *** *p* < 0.0005.

**Table 2 nutrients-16-01846-t002:** Positive answers to MEDAS questionnaire and adherence to the MedDiet.

	Whole Sample (*n* = 776)	Women (*n* = 553)	Men (*n* = 223)	*p*-Value
Olive oil, main dressing	749 (96.5%%)	540 (97.6%%)	209 (93.7%%)	^a^ 0.007 *
Olive oil, ≥4 ts/day	384 (49.5%%)	280 (50.6%%)	104 (46.6%%)	^a^ 0.31
Vegetables, ≥2 s/day	516 (66.5%)	402 (72.7%)	114 (51.1%)	^a^ 0.0001 ***
Fruits, ≥3 s/day	267 (34.4%)	187 (33.8%)	80 (35.9%)	^a^ 0.8
Red meat, <1 s/day	558 (71.9%)	412 (74.5%)	146 (65.5%)	^a^ 0.011 *
Butter, <1 s/day	636 (82.0%)	461 (83.4%)	175 (78.5%)	^a^ 0.109
Sweet beverage, <1 s/day	612 (78.9%)	438 (79.2%)	174 (78.1%)	^a^ 0.71
Wine, 7 s/week	90 (11.6%)	43 (7.8%)	47 (21.1%)	^a^ 0.0001 ***
Legumes, ≥3 s/week	343 (44.2%)	229 (41.4%)	114 (51.1%)	^a^ 0.014 *
Fish and seafood, ≥3 s/week	322 (41.5%)	223 (40.3%)	99 (44.4%)	^a^ 0.29
Sweets, <3 s/week	479 (61.7%)	333 (60.2%)	146 (65.5%)	^a^ 0.17
Nuts, ≥3/week	412 (53.1%)	302 (54.6%)	110 (49.3%)	^a^ 0.18
White meat over red	560 (72.2%)	420 (75.9%)	140 (62.8%)	^a^ 0.0001 ***
“Soffritto”	348 (44.8%)	248 (44.8%)	100 (44.8%)	^a^ 0.99
MEDAS score	8.08 ± 2.18	8.16 ± 2.08	7.88 ± 2.40	^b^ 0.108
Adherence to the MedDiet				^a^ 0.1
Low	84 (10.8%)	52 (9.4%)	32 (14.3%)	-
Medium	483 (62.2%)	346 (62.5%)	137 (61.4%)	-
High	209 (26.9%)	155 (28.1%)	54 (24.3%)	-

Positive answers to the MEDAS questionnaire. Compliance rates of at least 50% are indicated in italics. Data are expressed as numbers and percentages in parenthesis (*n* (%)) for categorical variables or as mean and standard deviation (M ± SD) for continuous variables. Vegetables daily serving: 1 medium portion = 200 g. Fruit daily serving: 1 serving = 100–150 g portion. Red meat/hamburgers/other meat daily serving: 1 medium portion = 100–150 g. Butter, margarine or cream daily serving: 1 medium portion = 12 g. Sweet or sugar-sweetened carbonated beverages daily serving: 1 medium portion = 200 mL. Wine daily serving: 1 medium portion = 125 mL. Legumes weekly serving: 1 portion = 150 g. Fish daily serving: 1 medium portion = 100–150 g. Seafood daily serving: 1 medium portion = 200 g. Nuts weekly serving: 1 portion of dairy product = 30 g. ^a^ A Chi-square test was performed to determine sex associations. ^b^ An independent samples non parametric Mann–Whitney U test was performed to determine sex differences. Abbreviations: MEDAS, Mediterranean diet adherence screener; MedDiet, Mediterranean diet; s, serving; ts, tablespoon. * *p* < 0.05, and *** *p* < 0.0005.

**Table 3 nutrients-16-01846-t003:** Binary and multinomial logistic regression analysis for COVID-19 symptoms and effects before and after vaccination cycle.

	COVID-19 Symptoms				Effect before Vaccination Cycle				Effect after Vaccination Cycle			
			95% CI for OR				95% CI for OR				95% CI for OR	
	*p*-Value	OR	Lower	Upper	*p*-Value	OR	Lower	Upper	*p*-Value	OR	Lower	Upper
MEDAS Score	0.001	3.130	2.232	4.388	0.545	1.220	0.952	1.098	0.831	1.008	0.937	1.085
Age	0.002	0.945	0.912	0.979	0.043	1.012	1.000	1.023	0.678	0.997	0.986	1.009
Sex	0.995	0.997	0.383	2.592	0.955	1.010	0.712	1.433	0.146	1.333	0.905	1.965
BMI	0.859	1.010	0.908	1.122	0.069	0.969	0.936	1.003	0.419	0.984	0.946	1.023
Lifestyle	0.033	2.602	1.082	6.253	0.936	1.013	0.732	1.402	0.923	0.983	0.692	1.397
Constant	0.016	0.025			0.043	3.006			0.047	3.268		

Abbreviations: BMI, Body Mass Index; MEDAS, Mediterranean diet adherence screener.

**Table 4 nutrients-16-01846-t004:** Comparison analysis between Mediterranean diet (MD) adherence and COVID-19 symptoms.

			MedDiet Adherence on Calculated MEDAS Score			Total
			Low	Medium	High	
COVID-19 symptoms	None	*n*	56	350	148	554
		%	66.7%	72.5%	70.8%	71.4%
	Asymptomatic	*n*	7	15	6	28
		%	8.3%	3.1%	2.9%	3.6%
	Mild	*n*	9	58	35	102
		%	10.7%	12.0%	16.7%	13.1%
	Flu	*n*	11	57	16	84
		%	13.1%	11.8%	7.7%	10.8%
	Pneumonia	*n*	1	3	4	8
		%	1.2%	0.6%	1.9%	1.0%
Total		*n*	84	483	209	776
		%	100.0%	100.0%	100.0%	100.0%

Chi Square = 14.26; *p* = 0.075. Abbreviations: MEDAS, Mediterranean diet adherence screener; MedDiet, Mediterranean diet.

**Table 5 nutrients-16-01846-t005:** Comparison analysis between COVID-19 symptoms and sex.

			Sex		Total
			Female	Male	
COVID-19 symptoms	None	*n*	392	162	554
		%	70.9%	72.6%	71.4%
	Asymptomatic	*n*	18	10	28
		%	3.3%	4.5%	3.6%
	Mild	*n*	77	25	102
		%	13.9%	11.2%	13.1%
	Flu	*n*	61	23	84
		%	11.0%	10.3%	10.8%
	Pneumonia	*n*	5	3	8
		%	0.9%	1.3%	1.0%
Total		*n*	553	223	776
		%	100.0%	100.0%	100.0%

Chi Square = 2; *p* = 0.736.

**Table 6 nutrients-16-01846-t006:** Associations between individual side effects and MEDAS groups.

Side Effects	Low	Medium	High	*p*-Value
At the beginning of the vaccine cycle				
Pain at the injection site	43	261	112	0.88
Tiredness	59	151	61	0.0001 ***
Fever and chills	21	122	45	0.58
Headache	11	105	40	0.056
Articular pains and muscle aches	19	119	47	0.81
Nausea	2	22	7	0.55
Enlarged lymph nodes	5	19	3	0.11
Diarrhea	0	11	4	0.37
Allergic reactions	1	4	6	0.1
At the end of the vaccine cycle				
Pain at the injection site	40	194	100	0.06
Tiredness	24	122	53	0.46
Fever and chills	22	91	42	0.55
Headache	17	85	41	0.91
Articular pains and muscle aches	17	80	30	0.23
Nausea	0	9	9	0.08
Enlarged lymph nodes	9	28	13	0.41
Diarrhea	2	6	2	0.65
Allergic reactions	2	3	5	0.19

*** *p* < 0.0005.

**Table 7 nutrients-16-01846-t007:** Associations between individual side effects and MEDAS groups among women sample.

Side Effect	Low	Medium	High	*p*-Value
At the beginning of the vaccine cycle				
Pain at the injection site	24	190	83	0.49
Tiredness	17	112	48	0.96
Fever and chills	14	89	33	0.54
Headache	6	80	31	0.15
Articular pains and muscle aches	11	90	37	0.71
Nausea	0	16	5	0.24
Enlarged lymph nodes	4	14	2	0.81
Diarrhea	0	7	4	0.51
Allergic reactions	1	2	4	0.16
At the end of the vaccine cycle				
Pain at the injection site	23	143	72	0.72
Tiredness	15	91	37	0.24
Fever and chills	14	64	32	0.62
Headache	10	60	30	0.93
Articular pains and muscle aches	11	62	22	0.16
Nausea	0	1	4	0.064
Enlarged lymph nodes	8	18	7	0.021 *
Diarrhea	0	4	1	0.56
Allergic reactions	1	2	3	0.45

* *p* < 0.05.

**Table 8 nutrients-16-01846-t008:** Associations between individual side effects and MEDAS groups among men sample.

Side Effect	Low	Medium	High	*p*-Value
At the beginning of the vaccine cycle				
Pain at the injection site	19	71	29	0.74
Tiredness	8	39	13	0.79
Fever and chills	7	33	12	0.94
Headache	5	25	9	0.92
Articular pains and muscle aches	8	29	10	0.77
Nausea	2	6	2	0.85
Enlarged lymph nodes	1	5	1	0.81
Diarrhea	0	4	0	0.27
Allergic reactions	0	2	2	0.4
At the end of the vaccine cycle				
Pain at the injection site	1	53	20	0.74
Tiredness	9	31	16	0.88
Fever and chills	8	27	10	0.78
Headache	7	25	11	0.98
Articular pains and muscle aches	6	18	8	0.95
Nausea	0	1	5	0.85
Enlarged lymph nodes	1	10	6	0.37
Diarrhea	2	2	1	0.37
Allergic reactions	1	1	2	0.41

## Data Availability

The data presented in this study are available on request from the corresponding author due to privacy.

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
