# Peer review of "Mediterranean Diet Influence on SARS-CoV-2 Vaccine Adverse Reaction: Friend or Foe?"

_nutrients, 2024, doi:10.3390/nu16121846_

Round 1

Reviewer 1 Report

Comments and Suggestions for Authors

This paper deals with an interesting topic, the influence of Mediterranean diet (MedDiet) on COVID and COVID vaccination. The title is “Mediterranean Diet influence on sex-specific SARS-CoV-2 vaccine responses”. Unfortunately, the paper fails to deliver any significant new information.

The mitigating effect of MedDiet on the severity of COVID symptoms was reported early on during the pandemic and has been confirmed in a large number of studies. In fact, the present paper in its Introduction and Discussion provides a good review of the topic.

This study is retrospective and uses a questionnaire on dietary habits and symptoms of COVID, and side effects after COVID vaccination. The study recruited by internet 553 women and 223 men, who answered questions by internet. This methodology is not adequate. The recruitment did not result in a representative sample of population. A retrospective survey is not suitable for recording of symptoms associated with COVID or vaccination. As for MedDiet, the survey may yield useful results, but that is not the point in the present study.

The findings of the MedDiet survey are shown in Table and Figures, which are redundant to each other and occupy most of the space in Results. In contrast, the results on the effect of MedDiet on COVID symptoms are presented in two short paragraphs and those on COVID vaccination is just one paragraph. This is totally disproportionate. The authors’ findings confirm that MedDiet is associated with decreased symptoms of COVID, but this is not a new finding.

MedDiet did not seem to reduce post-vaccination side effects. This is the main finding, but cannot be regarded as totally reliable because of the methodological problems.

In order to properly study the effect of MedDiet on “vaccine responses” (as the title claims), the study would have to be prospective and measure both immune responses and vaccine adverse effects, as is commonly done in clinical trials of vaccines.

With this retrospective study the least one could do is to lay out the results in full.

Comments on the Quality of English Language

minor editing

Author Response

Rome, May 28th, 2024

Dear Editor of Nutrients

First of all, my coauthors and I would like to thank You sincerely for this opportunity of cooperation, allowing us to resubmit our paper (ID: nutrients-3003315) after extensive revisions, for a possible publication upon “Nutrients”.

We profoundly thank the reviewers for the comments and useful suggestions aimed at improving the paper.

We thank You for your constructive critique and we hope the review process has led to an improved manuscript.

If additional changes are warranted, we will make them.

We hope that this revised version of our manuscript may now be found suitable for publication.

Sincerely,

Rossella Cianci, MD, PhD

This is a point-by-point list of changes made in the paper:

Reviewer 1

This paper deals with an interesting topic, the influence of Mediterranean diet (MedDiet) on COVID and COVID vaccination. The title is “Mediterranean Diet influence on sex-specific SARS-CoV-2 vaccine responses”. Unfortunately, the paper fails to deliver any significant new information.

The mitigating effect of MedDiet on the severity of COVID symptoms was reported early on during the pandemic and has been confirmed in a large number of studies. In fact, the present paper in its Introduction and Discussion provides a good review of the topic.

  • Thanks for your suggestion. It is well established the mitigating effect of MedDiet on the severity of COVID symptoms. The MedDiet is able to regulate gene expression related to oxidative stress; at the same time, sex per se exerts a regulation on gene expressions linked to inflammation and oxidative stress. To our knowledge, the novelty of our report is the association between being women and higher MedDiet adherence score with decreased COVID-19 symptoms and lymph node enlargement at the end of the vaccine cycle.

This study is retrospective and uses a questionnaire on dietary habits and symptoms of COVID, and side effects after COVID vaccination. The study recruited by internet 553 women and 223 men, who answered questions by internet. This methodology is not adequate. The recruitment did not result in a representative sample of population. A retrospective survey is not suitable for recording of symptoms associated with COVID or vaccination. As for MedDiet, the survey may yield useful results, but that is not the point in the present study.

  • The authors' aim was not to identify a representative sample of the entire population, but rather a sample that was representative of a specific age group of non-pathological and drug therapy-free subjects (<65 years old), otherwise it has been difficult to correlate the side effects found with the vaccine. This population routinely uses the PC and internet to work and more, thus a retrospective analysis by an online questionnaire, is appropriate to the aim. We did not underline these aspects sufficiently in the text; we therefore thank the Reviewer for his/her comments, and we have included these specific aspects among the objectives of the study in the revised manuscript.

The findings of the MedDiet survey are shown in Table and Figures, which are redundant to each other and occupy most of the space in Results. In contrast, the results on the effect of MedDiet on COVID symptoms are presented in two short paragraphs and those on COVID vaccination is just one paragraph. This is totally disproportionate. The authors’ findings confirm that MedDiet is associated with decreased symptoms of COVID, but this is not a new finding.

The authors thank for the Reviewers. Tables 4 and 5 have been added for analyses related to MEDAS and symptoms of COVID-19. Tables 6, 7, and 8 were added for analyses related to MEDAS and side effects, on the overall sample and divided by sex. In addition, Figures 1 and 2, have been removed and placed in Appendix B.

MedDiet did not seem to reduce post-vaccination side effects. This is the main finding, but cannot be regarded as totally reliable because of the methodological problems.

  • This observation was overcome because the methodological concern has been resolved: we have in fact uniquely defined the reference population.

In order to properly study the effect of MedDiet on “vaccine responses” (as the title claims), the study would have to be prospective and measure both immune responses and vaccine adverse effects, as is commonly done in clinical trials of vaccines.

With this retrospective study the least one could do is to lay out the results in full.

  • The authors thank the Reviewers. Tables 6, 7, and 8 show the analyses for MEDAS and side effects, on the overall sample and according to sex.

Reviewer 2 Report

Comments and Suggestions for Authors

This is an interesting research article with adequate novelty and quality. However, some points should be addressed.

- Subheadings should be added in the Abstract (e.g. Background, Methods, Results, and Conclusions).

- In line 27, the symbol "?" should be deleted.

- The 1st paragraph of the Introduction section should be incorporated to the 2nd paragraph.

- The 3rd paragraph concerning COVID-19 is too small and it should be increased by including more detailed information about the COVID-19 infection the the related inflammatory mechanisms.

- In line 64 the statement reporting ". Sex differences were also reported in infectious diseases, including COVID-19 [16]." should be enriched by adding more information about its topic.

- Concerning the Statistical Analysis section, it authors could be better to apply Mann-Whitney U test for non-normall distributed variables.

- The symbol "%" should be added in Tables when it is applicable.

- A comparison analysis for the data of Tables 1 and 2 should be performed, including in these tables the relevant p-values.

- The authors should report the confounding factors including the binary and multinomial logistic regression analyses.

- Relevant tables including the results of the binary and multinomial logistic regression analyses should be added in the Results section in which the confounding factors should be included.

- A table including a comparison analysis between the association of MD adherence with each COVID-19 symproms separately should be included.

- In addition, a table including a comparison analysis between the participants gender with each COVID-19 symproms separately should also be included either inot the full text or as a supplementary material.

- In the Discussion section, the statistical data into the brackets could be omitted.

- The sentence in lines 178-179 "Concerning the risk of SARS-CoV-2 infection and mortality, 178 women are known to be less affected than men [22,23]." needs more analysis based on the relevant references 22 and 23.

- Concerning the anti-inflammatory properties of Mediterranean Diet, the authors should provide more details after the line 190, reporting specific pro-inflammatory molecules whichw their levels could be decreased by adopting Mediterranean Diet.

- The sentence "Furthermore, MedDiet can prevent cardiovascular diseases and diabetes, which represent risk factors for severe COVID-19 [39]." in lines 201-203 needs a bit more analysis.

- In lines 235-236, does the statement "women reported a +0.67% incidence 235 of side effects compared with men..." is correct concerning the +0.67% incidence?

- The authors should include in the limitations of their study the retrospective study design which may strongly be related with recall biases as well as not causality associations.

- In lines 271-272, the sentence "A notable strength of our study lies in the unbiased selection of the sample population through various dissemination channels." needs more analusis and discussion to be more clear.

- In the conclusion section, the authors should provide some suggestions concerning what specific future studies could be performed based on the results of their study.

Comments on the Quality of English Language

Moderate editing of English language is recommended

Author Response

Rome, May 28th, 2024

Dear Editor of Nutrients

First of all, my coauthors and I would like to thank You sincerely for this opportunity of cooperation, allowing us to resubmit our paper (ID: nutrients-3003315) after extensive revisions, for a possible publication upon “Nutrients”.

We profoundly thank the reviewers for the comments and useful suggestions aimed at improving the paper.

We thank You for your constructive critique and we hope the review process has led to an improved manuscript.

If additional changes are warranted, we will make them.

We hope that this revised version of our manuscript may now be found suitable for publication.

Sincerely,

Rossella Cianci, MD, PhD

This is a point-by-point list of changes made in the paper:

Reviewer 2

This is an interesting research article with adequate novelty and quality. However, some points should be addressed.

- Subheadings should be added in the Abstract (e.g. Background, Methods, Results, and Conclusions).

- In line 27, the symbol "?" should be deleted.

                - The authors thank the Reviewer. The abstract was revised according to his/her comments.

- The 1st paragraph of the Introduction section should be incorporated to the 2nd paragraph.

- The authors thank the Reviewer. The paragraphs were revised according to his/her comments.

- The 3rd paragraph concerning COVID-19 is too small and it should be increased by including more detailed information about the COVID-19 infection the the related inflammatory mechanisms.

- The authors thank the Reviewer. We have added the following paragraph: After the binding between membrane angiotensin converting enzyme 2 (ACE 2) and SARS-CoV-2 spike S-glycoproteins, virus enter the cells and start immune response to the infection by an increase in proinflammatory cytokines, such as TNF-α, IL-6, IL-7, IL-8, Il-9, IL-10, Interferon gamma (IFN-γ), vascular endothelial growth factor (VEGF), and CRP, and elevated production of reactive oxygen species (ROS). SARS-CoV-2 can elicit a strong cytokine storm, a consequent dysregulation of immune system with the development of a systemic inflammation and multiorgan failure. During this inflammatory response, MedDiet could provide beneficial effects on COVID-19 outcomes, through its anti-inflammatory properties [8,14].

- In line 64 the statement reporting ". Sex differences were also reported in infectious diseases, including COVID-19 [16]." should be enriched by adding more information about its topic.

- The authors thank the Reviewer. The paragraph was revised, and the following text was included: “Indeed, women are more susceptible to autoimmune disorders and men are more prone to infectious diseases, potentially due to X chromosome protection and hormonal effects. Women often show stronger immune responses during viral infections, attributed to sex steroids, while higher ACE-2 expression in women may play a role in the less severe outcomes of COVID-19.”

- Concerning the Statistical Analysis section, it authors could be better to apply Mann-Whitney U test for non-normall distributed variables.

- The authors thank the Reviewer for his/her comments. This is a typo, because the authors know well that, in the presence of non-normally distributed data, non-parametric tests must be applied. Therefore, the sentence in the revised version of the manuscript has been modified as follows: “Differences in the MEDAS score variable between women and men were determined using Mann-Whitney U-test for independent samples” (see page 3 rows 107-108). In addition, the following sentence at page 5 row 140 was modified as follows: “A Mann-Whitney U-test for independent samples reported a significant difference on MEDAS score between those who have or have not contracted COVID-19 (p < 0.0001)”.

- The symbol "%" should be added in Tables when it is applicable.

- The authors thank the Reviewer. The Tables were revised according to his/her comments.

- A comparison analysis for the data of Tables 1 and 2 should be performed, including in these tables the relevant p-values.

- The authors thank the Reviewer. A comparison analysis for the data in Tables 1 and 2 was conducted, including the relevant p-values. For this purpose, the Mann-Whitney U test for continuous variables and the Chi-square test for categorical variables were performed. In addition, the following text was included in the Results section: “Significant sex differences were observed in Weight (p<0.0001), Height (p<0.0001), and BMI (p<0.0001). No other sex differences were reported. […..] Significant sex differences were observed in Olive Oil (p<0.007), Vegetables (p<0.0001), Red Meat (p<0.011), Wine (p<0.0001), Legumes (p<0.014) and White Meat (p<0.0001) consumption. No other sex differences were reported.”

- The authors should report the confounding factors including the binary and multinomial logistic regression analyses.

- The authors thank the Reviewer. In the logistic regression analysis, the following confounding factors were included: age, gender, BMI and lifestyle (sedentary/active). This sentence was also added in the revised version of the manuscript (see page 3 rows 113-114).

- Relevant tables including the results of the binary and multinomial logistic regression analyses should be added in the Results section in which the confounding factors should be included.

- Authors thank the Reviewer for his/her comments. A new table (Table 3) including the results of the binary and multinomial logistic regression analyses was included.

- A table including a comparison analysis between the association of MD adherence with each COVID-19 symproms separately should be included.

- Authors thank the Reviewer for his/her comments. A new table (Table 4) including a comparison analysis between the association of MD adherence and each COVID-19 symptom was included.

- In addition, a table including a comparison analysis between the participants gender with each COVID-19 symproms separately should also be included either inot the full text or as a supplementary material.

- Authors thank the Reviewer for his/her comments. A new table (Table 5) including the comparison analysis between the gender of participants and each COVID-19 symptom was included.

- In the Discussion section, the statistical data into the brackets could be omitted.

-              The authors thank the Reviewer. The Discussion was revised according to his/her comments.

- The sentence in lines 178-179 "Concerning the risk of SARS-CoV-2 infection and mortality, 178 women are known to be less affected than men [22,23]." needs more analysis based on the relevant references 22 and 23.

- The authors thank the Reviewer. The paragraph was revised, and the following text was included: “Men with COVID-19 are known to have higher mortality rates than women, possibly due to higher rates of comorbidities related to male norms, such as smoking and alcohol consumption. In addition, men have also demonstrated prolonged viral shedding, lower cure rates, and greater disease severity likely influenced by hormonal factors and ACE-2 expression in the testis. While susceptibility to disease does not differ significantly between genders, biological factors such as innate immunity and hormone levels may contribute to these disparitie.”

- Concerning the anti-inflammatory properties of Mediterranean Diet, the authors should provide more details after the line 190, reporting specific pro-inflammatory molecules whichw their levels could be decreased by adopting Mediterranean Diet.

- The authors thank the Reviewer. The paragraph was revised, and the following text was included: “It is well known that MedDiet leads to a significant reduction in circulating markers of oxidative stress and inflammation, such as ox-LDL and CRP levels, and significant down-regulation of pro-inflammatory genes such as CCL5, UCP2, BCL2, IRAK1 and DUOX2, compared with a Western diet.”

- The sentence "Furthermore, MedDiet can prevent cardiovascular diseases and diabetes, which represent risk factors for severe COVID-19 [39]." in lines 201-203 needs a bit more analysis.

- The authors thank the Reviewer. The paragraph was revised, as follows: “MedDiet diet, renowned for its preventive effects on cardiovascular disease and type 2 diabetes, offers potential benefits in mitigating the severity of COVID-19 infection. Rich in plant-based components, MedDiet provides bioactive polyphenols. Particular, flavonoids are known for their antioxidant and anti-inflammatory properties and can counteract the exaggerated inflammatory response associated with severe COVID-19 disease, potentially improving outcomes. In addition, MedDiet components, such as nuts and dried fruits, offer antioxidant and anti-inflammatory molecules that promote cardiometabolic health.”

- In lines 235-236, does the statement "women reported a +0.67% incidence 235 of side effects compared with men..." is correct concerning the +0.67% incidence?

- The authors thank the Reviewer. The sentence was revised according to his/her comment, as follows: “In our study, women reported a rate of +0.67% side effects compared with men at the beginning of the vaccination cycle.”

- The authors should include in the limitations of their study the retrospective study design which may strongly be related with recall biases as well as not causality associations.

- The authors thank the Reviewer. We have added the sentence to our limitations: “In addition, due to the retrospective study design the recall biases as well as not causality associations cannot be excluded.”

- In lines 271-272, the sentence "A notable strength of our study lies in the unbiased selection of the sample population through various dissemination channels." needs more analusis and discussion to be more clear.

- The authors thank the Reviewer. The paragraph was revised, and the following text was included: “A notable strength of our study lies in the unbiased selection of the sample population facilitated by the widespread adoption of digital channels in Italy. With over 50 million internet users and nearly 44 million social media users in 2023, representing 86.1% and 74.5% of the Italian population, respectively, digital platforms provided a broad and accessible means of reaching potential participants. Additionally, the presence of 78.19 million cellular mobile connections, exceeding the Italian population, highlights the extensive use of mobile technology, further enhancing the reach and inclusivity of our recruitment efforts.”

- In the conclusion section, the authors should provide some suggestions concerning what specific future studies could be performed based on the results of their study.

- The authors thank the Reviewer. The Conclusions section was revised, and the following text was included: “Based on our evidence, future studies could explore sex differences in the long-term effects of MedDiet adherence on COVID-19 outcomes and vaccination responses in different populations. Simultaneously, clinical trial studies should be conducted to investigate the underlying mechanisms by which MedDiet components modulate immune responses to viral infections and vaccination, with a focus on sex-specific differences. These investigations will provide valuable insights into the potential of MedDiet interventions in enhancing vaccine responses and reducing the severity of COVID-19. In addition, expanding research efforts to include larger and more diverse populations beyond Italy through population-based studies will clarify the generalizability of findings and consider cultural and dietary variations.”

Round 2

Reviewer 1 Report

Comments and Suggestions for Authors

The revised article is acceptable but the title needs to be changed. I propose the following: "Lack of Mediterranean diet influence on SARS-CoV-2 vaccine adverse reactions"

Reviewer 2 Report

Comments and Suggestions for Authors

I would like to thank the authors for their trust to my comments and suggestions. The authors have ssignificantly improved their quality, increasing its quality presentantion and its scientific soundness.

Comments on the Quality of English Language

Minor editing of English language required